# Untwining Anti-Tumor and Immunosuppressive Effects of JAK Inhibitors—A Strategy for Hematological Malignancies?

**DOI:** 10.3390/cancers13112611

**Published:** 2021-05-26

**Authors:** Klara Klein, Dagmar Stoiber, Veronika Sexl, Agnieszka Witalisz-Siepracka

**Affiliations:** 1Department of Biomedical Science, Institute of Pharmacology and Toxicology, University of Veterinary Medicine Vienna, 1210 Vienna, Austria; klara.klein@vetmeduni.ac.at (K.K.); veronika.sexl@vetmeduni.ac.at (V.S.); 2Department of Pharmacology, Physiology and Microbiology, Division Pharmacology, Karl Landsteiner University of Health Sciences, 3500 Krems, Austria; dagmar.stoiber@kl.ac.at

**Keywords:** JAK, STAT, JAK inhibitor, NK cells, T cells, leukemia

## Abstract

**Simple Summary:**

The Janus kinase-signal transducer and activator of transcription (JAK-STAT) pathway is aberrantly activated in many malignancies. Inhibition of this pathway via JAK inhibitors (JAKinibs) is therefore an attractive therapeutic strategy underlined by Ruxolitinib (JAK1/2 inhibitor) being approved for the treatment of myeloproliferative neoplasms. As a consequence of the crucial role of the JAK-STAT pathway in the regulation of immune responses, inhibition of JAKs suppresses the immune system. This review article provides a thorough overview of the current knowledge on JAKinibs’ effects on immune cells in the context of hematological malignancies. We also discuss the potential use of JAKinibs for the treatment of diseases in which lymphocytes are the source of the malignancy.

**Abstract:**

The Janus kinase-signal transducer and activator of transcription (JAK-STAT) pathway propagates signals from a variety of cytokines, contributing to cellular responses in health and disease. Gain of function mutations in JAKs or STATs are associated with malignancies, with *JAK2**^V617F^* being the main driver mutation in myeloproliferative neoplasms (MPN). Therefore, inhibition of this pathway is an attractive therapeutic strategy for different types of cancer. Numerous JAK inhibitors (JAKinibs) have entered clinical trials, including the JAK1/2 inhibitor Ruxolitinib approved for the treatment of MPN. Importantly, loss of function mutations in JAK-STAT members are a cause of immune suppression or deficiencies. MPN patients undergoing Ruxolitinib treatment are more susceptible to infections and secondary malignancies. This highlights the suppressive effects of JAKinibs on immune responses, which renders them successful in the treatment of autoimmune diseases but potentially detrimental for cancer patients. Here, we review the current knowledge on the effects of JAKinibs on immune cells in the context of hematological malignancies. Furthermore, we discuss the potential use of JAKinibs for the treatment of diseases in which lymphocytes are the source of malignancies. In summary, this review underlines the necessity of a robust immune profiling to provide the best benefit for JAKinib-treated patients.

## 1. JAK-STAT Signaling in Hematological Malignancies 

The Janus kinase-signal transducer and activator of transcription (JAK-STAT) pathway propagates signals downstream of growth factor, hormone, and cytokine receptors. Thereby the JAK-STAT pathway regulates development, survival, proliferation, differentiation, and functionality of cells within the immune system [1,2,3,4,5].

The four JAKs, JAK1-3, and TYK2, are non-receptor tyrosine kinases which associate with their respective receptors. While some receptors associate with a pair of a specific JAK-family member, others are linked to more than one JAK. Ligand binding induces oligomerization of transmembrane receptors and activation of JAKs by auto- and/or trans-phosphorylation events. Subsequently, JAKs phosphorylate tyrosine residues on the receptors representing docking sites for STATs (STAT1-6) via their Src homology 2 (SH2) domain. Upon phosphorylation, STATs undergo a conformation change and switch from an anti-parallel to a parallel homo/heterodimer and translocate to the nucleus to regulate gene expression [2,3,5,6,7,8,9]. Additional posttranslational modification, formation of multimeric complexes with co-factors, and epigenetic remodeling processes add to the complexity of the transcriptional regulation initiated by the JAK-STAT signaling pathway and allow for its diverse functional consequences [5,6,10]. JAK1 is key for interferon (IFN) signaling, in combination with JAK2 in case of IFN-γ and with TYK2 in case of type I and type III IFN signaling. Together with JAK3, which is directly associated with the common γ chain (γc) of cytokine receptors, JAK1 signals downstream of the γc-dependent cytokines interleukin (IL)-2, IL-4, IL-7, IL-9, IL-15, and IL-21. JAK2 is responsible for signaling downstream of erythropoietin (EPO), thrombopoietin (TPO), growth hormone, prolactin, leptin, IL-3, IL-5, and granulocyte-macrophage colony-stimulating factor (GM-CSF). JAK1, JAK2, and, in some cases, TYK2 are activated by gp130 cytokine family members (IL-6, IL-11, IL-31, oncostatin M (OSM), ciliary neurotrophic factor (CNTF), leukemia inhibitor factor (LIF), cardiotropin-1 (CT-1), neurotrophin-1 (NNT-1)) and IL-13. TYK2 is involved in signaling of IL-10-like cytokines (IL-10, IL-19, IL-20, IL- 22, IL-24, and IL-26) in conjunction with JAK1 and mediates IL-12/IL-23 cytokine signaling together with JAK2 [8,11,12]. 

Deregulated JAK-STAT signaling in the hematopoietic system is associated with a wide range of alterations, including immunodeficiency, autoimmunity, and transformation [4,5,13,14]. The discovery of hyperactivation of JAK-STAT pathway members in inflammatory pathologies (such as rheumatoid arthritis (RA), psoriasis, and inflammatory bowel disease) and hematological cancers triggered the development of JAK inhibitors (JAKinibs) as potential treatment options [4,15,16]. In cancer, aberrant activation of the JAK-STAT pathway is achieved by different means, including deregulated upstream signals or feedback loops, gene amplifications, generation of fusion proteins as well as gain-of-function (GOF) mutations [4,17]. The latter will be discussed in more detail below focusing on hematological malignancies. Dysregulated JAK-STAT signaling also plays a role in solid cancers, as reviewed before [4,18,19].

An overview of GOF mutations in the JAK-STAT signaling pathway occuring in hematological malignancies is presented in Table 1. JAKs consist of multiple domains, including an N-terminal FERM domain and an SH2-like domain that promote receptor interaction, a pseudokinase domain (JH2) with regulatory capacity, and the C-terminal catalytically active kinase domain (JH1) [4,20,21,22,23,24]. GOF mutations in JAKs are frequently located in the JH2 domain, where they lead to an altered auto-inhibitory function and render JAKs constitutively active [22,25]. The most prominent example is the somatic *JAK2^V617F^* mutation that is highly prevalent in myeloproliferative neoplasms (MPNs). MPNs, including polycythemia vera (PV), essential thrombocythemia (ET) and primary myelofibrosis (PMF), are a group of diseases in which myeloid precursors produce increased numbers of blood cells with the potential to evolve into acute myeloid leukemia (AML) [26,27,28,29,30,31,32,33,34,35]. The equivalent germline *JAK2^V617I^* mutation, besides other pseudokinase domain mutations, has been described in hereditary essential thrombocytosis [36,37,38]. The corresponding somatic JAK1 JH2 mutations (V658I or V658F) are present in patients with acute lymphoblastic leukemia (ALL), T-cell prolymphocytic leukemia (T-PLL), and myeloid malignancies [4,15,25,39,40,41,42]. Somatic JH2 mutations are also found in JAK3 (e.g., A572V and A573V) in T-cell malignancies (T-PLL, T-ALL, early T-cell precursor ALL (ETP-ALL)), AML, and malignancies of natural killer (NK) cell origin (e.g., extranodal NK/T-cell lymphoma (NKTCL)) [4,41,43,44,45,46,47,48,49,50,51,52,53]. Rare cases of somatic and germline JH2 mutations in TYK2 have been described that are associated with ALL [4,54,55,56]. Other activating JAK mutations associated with hematological malignancies are found in the FERM, linker, and JH1 domain [4,15,32,57]. 

GOF mutations in STATs have also been described in hematological malignancies [5,10,58] and most frequently occur in the SH2 domain, which is required for dimerization and transcriptional activity. SH2 domain mutations of STAT3 (e.g., Y640F, D661Y) and STAT5B (e.g. N642H, Y665F/H) are found in mature T-cell and NK-cell neoplasms (see Table 1 for details of cancer subtypes), T-ALL, diffuse large B-cell lymphoma (DLBCL) and myeloid diseases (eosinophila, AML, chronic neutrophilic leukemia) [59,60,61,62]. Activating STAT mutations do also exist outside of the SH2 domain in hematological cancers [60,63,64,65] and DNA binding domain mutations of STAT6 are associated with B-cell malignancies [66,67,68,69].

The identification of *JAK2^V617F^* as a driver mutation in the majority of PV cases and in over 50% of ET and PMF cases was a major breakthrough and provided the rationale for the development of JAKinibs [26,27,28,70]. Ruxolitinib (JAK1/2 inhibitor) has been approved by the US Food and Drug Administration (FDA) and European Medicines Agency (EMA) for the treatment of intermediate and high risk PMF and secondary MF and second line for PV patients resistant or intolerant to hydroxyurea [71]. JAKinib treatment prolongs overall survival and reduces splenomegaly typically associated with MPNs, but does not induce complete remission. Importantly, Ruxolitinib triggers comparable therapeutic responses in MPN patients irrespective of the *JAK2^V617F^* mutation. Patients harboring deletions in the calreticulin gene (CALR) or mutations in the MPL receptor also show therapeutic responses as the constitutive activation of the JAK-STAT pathway represents a common feature of MPNs. As a consequence, JAKinibs are extensively tested in (pre)clinics as therapies for a variety of malignancies with overactivation of the JAK-STAT pathway due to mutations (see Table 1) or due to constitutive activation of the pathway. As such, diseases driven by the *STAT5B^N642H^* mutation are partially responsive to Ruxolitinib [72,73]. Table 2 summarizes ongoing clinical trials for JAKinib treatment of hematological malignancies. 

Activation of the JAK-STAT pathway by mutations mostly increases sensitivity to JAKinibs (as exemplified in Table 1), but no JAKinib is available that specifically targets a mutated allele. Such inhibitors may circumvent potential side-effects of current JAKinibs that also hit non-transformed cells [4]. 

## 2. JAK-STAT Signaling in the Immune System 

The JAK-STAT pathway has a critical role in the development and function of immune cells. Loss-of-function (LOF) mutations in JAKs or STATs are associated with immune deficiencies or enhanced susceptibility to infections [5]. The most severe phenotype occurs in patients with LOF in signaling components downstream of γc-dependent cytokines, namely JAK3 and STAT5B, which manifest as severe combined immune deficiencies (SCIDs) [4,110]. The severe immune deficiency is predominantly linked to impaired IL-7 and IL-15 signaling [14]. Interestingly, STAT5B LOF mutations may also provoke autoimmune phenotypes due to non-functional T regulatory cells (Tregs) [111]. LOF mutations in STAT1 and TYK2 enhance the susceptibility to bacterial and viral infections [110,112], while LOF in STAT2 and STAT4 increase the incidence of viral and fungal infections, respectively [113,114,115,116]. Similarly, GOF mutations of STAT1 and STAT3 may suppress the immune system and manifest with impaired anti-viral, anti-bacterial, and for STAT1, also anti-fungal responses [110,117]. Knockout mice for JAKs and STATs generally recapitulate the phenotypes of patients with LOF mutations and provide a valuable tool to understand the molecular mechanisms of JAK-STAT signaling in diseases. The consequences of JAK and STAT knockouts in mice and LOF mutations in humans have been reviewed extensively and the most prominent immune phenotypes are summarized in Table 3. The availability of conditional knockouts provides the opportunity to overcome embryonic and perinatal lethality of JAK1, JAK2, STAT3, or STAT5A/B-deficient mice. Conditional knockouts for STAT3 and STAT5A/B allowed to uncover the effects of STAT3 and STAT5 in T-cell differentiation and memory, dendritic cell function, and NK-cell tumor surveillance (reviewed in [5]). Conditional JAK knockouts have not yet been studied in depth despite the severe immune defects of knockout mice. Inducible deletion of JAK1 markedly impairs stem cell homeostasis and reduces the frequencies of B cells [118]. Similarly, NK cell-specific deletion of JAK1 almost completely abrogates the presence of peripheral NK cells [119]. In contrast, inducible deletion of JAK2 fails to impact on lymphopoiesis [120] and NK cell-specific JAK2 deficiency does not interfere with NK-cell homeostasis [119]. JAK2-deficient T cells are skewed towards Th2 and Treg polarization resulting in reduced graft versus host disease (GvHD) [121]. In addition, NK cell-specific deletion of TYK2 decreases anti-bacterial responses while leaving NK cell-mediated tumor surveillance intact in contrast to global TYK2 knockout mice [122,123,124]. These studies highlight the importance of the JAK-STAT pathway for immune responses and point towards a potential risk for patients when drug-targeting the JAK-STAT pathway.

## 3. The Effects of JAKinibs on the Immune System: Focus on Cytotoxic Lymphocytes 

JAKinibs provide a powerful tool in targeted cancer therapy. In light of the above-described immune alterations, JAKinibs are expected to provoke immune suppression. We here provide an overview of currently available JAKinibs undergoing clinical trials for treatment of hematological diseases and focus on their immune effects. 

### 3.1. Ruxolitinib

Ruxolitinib (Jakavi^®^) successfully reduces the systemic symptoms of MPN patients [71] while it does not eliminate the malignant clone [165]. MPNs are characterized by the increased levels of pro-inflammatory cytokines that are associated with enhanced NF-κB signaling [166]. The enhanced cytokine production contributes to systemic symptoms and is of prognostic value in MF [167]. Malignant and non-malignant cells are the source of proinflammatory cytokines and the success of Ruxolitinib treatment depends on the suppression of chronic inflammation [168]. Thus, MPN requires Ruxolitinib treatment for a prolonged period, which also exerts immune suppressive effects [71]. A recent approval of Ruxolitinib for treatment of steroid-refractory acute GvHD underlines its potent repressive activity on the immune system [169]. In line, the 5-year follow-up study of the COMFORT-II clinical trial uncovered that Ruxolitinib-treated MPN patients manifest with neutropenia and leukopenia, urinary tract infections, pneumonia, sepsis, tuberculosis, and herpes zoster infections [170]. In the majority of case reports, severe viral or bacterial infections forced the discontinuation of the therapy (reviewed in [171]). A recent retrospective study failed to identify a significantly increased risk of infections in Ruxolitinib-treated MPN patients which may reflect the low number and heterogeneity of patients in the control group [172]. Porpaczy et al., reported a strongly increased risk for Ruxolitinib-treated patients to develop B-cell lymphoma [173]; this result is in contrast to other studies that reported an increased risk of skin cancer upon Ruxolitinib treatment [174,175,176]. Of note, lymphoma is a serious rare adverse event in RA patients treated with another JAK1/2 inhibitor Baricitinib [177]. It still remains to be determined whether there is an increased rate of infections or secondary malignancies in large patient cohorts, which is challenging for rare diseases such as MPNs.

The basic analysis of immune functions in several cell types clearly illustrates the immune suppressive effects of Ruxolitinib. NK cells are the first-line of defense against virally infected and transformed cells and crucially depend on IL-15/JAK1/3/STAT5 signaling for survival and function. Ruxolitinib-treated patients harbor decreased NK-cell numbers with an immature phenotype. In addition, in vitro Ruxolitinib treatment of healthy donor-derived NK cells impairs their cytotoxic activity [178]. Furthermore, Ruxolitinib interferes with the dendritic cell (DC)—NK cell interaction which is required for full blown activation [179]. This effect is linked to an impaired NK cell-intrinsic IL-15 signaling upon DC trans-presentation [180]. In addition, Ruxolitinib inhibits DC development and activation in vitro and in vivo and is also associated with an impaired DC migration and clearance of adenoviral infections in mice [181]. The reduced migration ability of DCs is caused by the off-target inhibition of Rho-associated coiled-coil kinases [182]. 

The suppressive effects of Ruxolitinib extend to adaptive immune cells. Three independent studies showed significantly reduced Treg numbers in Ruxolitinib-treated patients, an effect that occurs rapidly and is long-lasting [183,184,185]. Effects on other T-cell subtypes are less clear; reduced Th1 and Th17 cell numbers were observed in patients upon three weeks of treatment whereas another patient cohort failed to show any difference in total CD4+ or Th1 cell numbers after a three- or six-months treatment period [183,184]. In contrast, in a murine model of GvHD, Th1 cells were reduced and Treg numbers were increased in the Ruxolitinib-treated group [186]. In any case, T-cell activity is significantly impaired by Ruxolitinib via interference with the IL-2/JAK1/3/STAT5 pathway [183]. Opposing effects were observed for B cells, where Ruxolitinib treatment normalized the decreased numbers of B cells in MPN patients to physiological levels. The underlying mechanism remains enigmatic [187].

### 3.2. Tofacitinib

Tofacitinib (CP-690,550) is a JAK1/3 inhibitor with a lower affinity for JAK2 and is clinically used for the treatment of RA, psoriatic arthritis, and ulcerative colitis due to its immunosuppressive effects [16,188,189,190]. The immunosuppressive activity of Tofacitinib that is exploited in autoimmune diseases is linked to its ability to block signaling downstream of multiple cytokines. This results in a suppressed differentiation of pathogenic Th1 and Th17 cells and reduced pro-inflammatory signals by innate immune cells [191,192,193,194]. In animal models, Tofacitinib treatment reduces CD8+ T and NK-cell numbers and affects NK-cell differentiation [195,196,197,198], which is discussed to be the consequence of blocking γc-utilizing cytokines that are essential for lymphocyte survival and maturation [199,200,201]. Decreased lymphocyte numbers are associated with immunosuppression and prolong allograft survival and reduce delayed-type hypersensitivity (DTH), while impairing anti-tumor responses in an experimental lung metastasis mouse model of colon cancer [195,196,197,198,202]. In RA patients, long-term Tofacitinib treatment gradually decreased absolute lymphocyte (ALC) and T-cell counts. NK-cell numbers drop in a dose-dependent manner upon short-term Tofacitinib treatment, while they increase upon mid- to long-term treatment. The latter is discussed to be the consequence of increased IL-15 availability [194,203,204,205,206]. Of note, changes in lymphocyte numbers are reversible upon treatment withdrawal [194,205,206,207].

The reduced NK and T-cell numbers are associated with an only transient impairment of T-cell-mediated responses while NK-cell functions are consistently impaired even after treatment withdrawal in healthy volunteers [206]. Tofacitinib does not interfere with the viability of healthy donor NK cells but decreases IL-2-mediated NK-cell activation at concentrations used in RA patients. NK cells display an impaired NK-cell receptor expression, degranulation, cytotoxicity, and cytokine production [208]. In addition, Tofacitinib decreases expression levels of CD80/CD86 and thereby the T-cell stimulatory capacity of DCs through suppression of type I IFN signaling [209]. In contrast to effector T cells, Tregs appear less sensitive to Tofacitinib, as their numbers and suppressive activity are preserved upon treatment [206,210,211]. Similar to Ruxolitinib, B-cell numbers increase upon Tofacitinib treatment [194,205,207]. Despite a potential effect on B-cell maturation [212], humoral responses to vaccines are not affected by Tofacitinib, indicating a neglectable suppressive effect on B-cell functions [194,205,206,207]. 

The suppressive effects on T and NK cells translate into adverse effects of Tofacitinib including higher incidences of upper respiratory tract infections, pneumonia, and herpes zoster [194]. Initial reports did not show higher frequencies of lymphoma development in Tofacitinib-treated RA patients, but a slight increase in Epstein–Barr virus (EBV) association [208,213]. More recently, preliminary results from a safety clinical trial (NCT02092467) revealed an increased risk of malignancies, especially lung cancer, upon Tofacitinib treatment compared to treatment with tumor necrosis factor (TNF) inhibitors [214].

### 3.3. Fedratinib

Fedratinib (SAR-302503, TG-101348) has recently been approved for the treatment of intermediate-2 or high-risk primary or secondary MF. Fedratinib has a high specificity for JAK2 when compared to Ruxolitinib [215]. In addition, Fedratinib inhibits fms like tyrosine kinase 3 (FLT3) and bromodomain-containing protein 4 (BRD4). While the role of FLT3 inhibition in the therapeutic outcome is unclear, inhibition of BRD4 strongly diminishes NF-κB-driven pro-inflammatory cytokines which contribute to MPN pathology [166]. The anti-inflammatory effects of Fedratinib are comparable to the effects of Ruxolitinib but were assigned to BRD4 blockade rather than JAK1 inhibition [11]. The specificity of Fedratinib towards JAK2 predicts a less immune suppressive potential but clinical data on the risk of infections in large patient cohorts are still lacking. Clinical trials point at neutropenia and slightly higher frequencies of urinary tract infections upon Fedratinib treatment [215].

While it is well established that Fedratinib diminishes inflammation in MPN patients the exact effects on each (immune) cell type remains to be determined. NK-cell proliferation and function are less affected by Fedratinib than by Ruxolitinib. Although Fedratinib impairs IL-2 and soluble IL-15-mediated activation of STAT5 in NK cells, it does not interfere with the DC-dependent trans-presentation of IL-15 to NK cells [179]. It is attractive to speculate that this mechanism overrides any other effect in vivo rendering Fedratinib significantly less suppressive for NK cells. Of note, Fedratinib inhibits NK-cell development in vitro in a suppressor of cytokine signaling protein 2 (SOCS2)-deficient background by an unknown mechanism [216].

In vitro treatment of healthy donor peripheral blood mononuclear cells (PBMCs) with Fedratinib impairs DC maturation but retains the ability to induce T-cell proliferation which is in contrast to Ruxolitinib. JAK2 inhibition fails to block the IL-2/STAT5 pathway which accounts for the suppressive effect of Ruxolitinib on T cells [217]. Despite the fact that in vitro Treg numbers remain unaffected by Fedratinib [179], in vivo, a few patients react with decreased Treg numbers to Fedratinib treatment [184]. A more detailed analysis on the effects of Fedratinib for immune functions may be expected from the upcoming phase 4 trials.

### 3.4. Momelotinib

Momelotinib (CYT387) is a JAK1/2 inhibitor that is currently in phase 3 clinical trials for MF and shows equal therapeutic effects to Ruxolitinib in JAKinib-naive patients. The major advantage of Momelotinib is the fact that it induces anemia to a reduced extent [218,219]. The effects of Momelotinib on the immune system remain to be determined. The only information available so far is the comparable impairment of T-cell subsets upon in vitro treatment with Momelotinib and Ruxolitinib [220], which was expected as both drugs target the same spectrum of cytokines.

### 3.5. Pacritinib

Pacritinib (SB1518) is an inhibitor selective for JAK2, FLT3, interleukin receptor-associated kinase (IRAK), and Colony stimulating factor 1 receptor (CSF1R) and is currently undergoing phase 2/3 clinical trials for MPNs [221]. Pre-clinical data identified that AML cells are sensitive to Pacritinib due to its ability to inhibit IRAK signaling thereby providing a rationale for its use in AML irrespective of the JAK2 or FLT3 mutation status [222]. So far, only limited information regarding the risk for serious infections is available from clinical trials and only little data exist on its effects on immune cell subsets. In the context of GvHD, Pacritinib does not affect non-alloreactive T cells and Tregs which contrasts observations made with Ruxolitinib [121]. Confirmation of beneficial effects of Pacritinib in GvHD stems from a phase 1 clinical trial where combination of Pacritinib with mTOR inhibitors successfully limited acute GvHD [223]. Unexpectedly, Pacritinib has a pronounced effect on NK-cell activity in vitro [121], which is not observed for the JAK2/FLT3-specific inhibitor Fedratinib. The NK-cell effect of Pacritinib is thus most likely not related to JAK2. Preliminary data from our laboratory show that unlike Ruxolitinib, Pacritinib impairs cytotoxic activity of murine and human NK cells already upon short-term exposure. Long-term exposure abolishes this difference; NK-cell functions are impaired to the same extent irrespective of the inhibitor used. The effect of Pacritinib is not related to JAK2, as an impairment of cytotoxicity was also observed in JAK2-deficient NK cells. The nature of this off-target effect is currently unclear. Betts et al., speculated that Pacritinib affects NK cells by interfering with TYK2 [121]. This appears unlikely in view of the fact that TYK2 deficiency only in NK cells does not impair cytotoxicity [122]. 

### 3.6. Itacitinib

Itacitinib (INCB039110) is a potent JAK1-specific inhibitor with pronounced anti-inflammatory activity in preclinical disease models. It is currently undergoing clinical trials for treatment of autoimmune/inflammatory diseases and cancers [207,224]. Clinical trials in hematological malignancies are summarized in Table 2. In contrast to first generation pan-/multiple-JAK inhibitors, this JAK1-specific inhibitor holds the promise of dampening inflammation driven by JAK1-dependent cytokines (like IFN-γ, IL-6), while having reduced side effects [207,224,225,226]. Itacitinib efficiently inhibits IL-2-induced phosphorylation of STAT3 and STAT5 and thereby IL-2-mediated T-cell proliferation. It suppresses pro-inflammatory cytokine signaling, including IL-6 signaling, and IL-17 production by T cells upon IL-23 stimulation [224]. These anti-inflammatory and immunosuppressive effects were confirmed in vivo in rodent models of experimentally induced arthritis, inflammatory bowel disease and alloreactive inflammatory acute GvHD [224,227]. In an MHC-mismatched mouse model of acute GvHD, Itacitinib treatment reduced inflammatory cytokine levels (including IFN-γ, TNF-α, IL-6, IL-13, and IL-1β) and T-cell numbers in the inflamed colon tissue. In contrast, T-cell numbers in blood and spleen remained unaffected. Interestingly, graft-versus-leukemia responses in this mouse model were preserved upon Itacitinib treatment [227,228]. While Itacitinib treatment did not drastically affect Tregs numbers, it was associated with a decreased level of activated HLA-DR+CD38+ Tregs [228].

Preliminary data point towards good clinical efficacy of Itacitinib in autoimmune diseases [207,229,230,231,232], but no significant effect was observed in a trial for GvHD compared to corticosteroids (NCT03139604) [224]. Of note, the clinical potential of Itacitinib for prevention of cytokine release syndrome in patients treated with anti-CD19 CAR-T cells is currently being investigated (NCT04071366) [226]. Itacitinib is also explored for its anti-tumor activities in MPNs and B-cell malignancies as mono- or combination-therapy (Table 2). The potential and efficacy of JAK1-specific inhibitors in comparison to first-generation JAKinibs regarding their anti-tumor, anti-inflammatory, or immunosuppressive effects in different disease entities still awaits further evaluation.

## 4. When an Immune Cell Becomes Cancerous—Hijacking JAKinibs’ Immunosuppressive Side Effects for Treatment of NK/T-Cell Tumors

The strong suppressive effects of Ruxolitnib on NK and T cells are now exploited for the treatment of hematological diseases originating from cytotoxic lymphocytes including NKTCL, aggressive NK-cell leukemia (ANKL), and T-cell lymphomas/leukemias. NKTCL and ANKL are aggressive diseases with poor prognosis. They harbor alterations of the JAK-STAT signaling pathway, including STAT3 mutations and mutations in epigenetic modifiers [86,91,100]. Drug screens revealed a synergistic activity of the JAK1/2 inhibitor Ruxolitinib in combination with the BCL2 inhibitor Venetoclax in NKTCL and ANKL cell lines [100]. A synergistic efficacy of Ruxolitinib and BCL2 inhibitors was also shown in T-cell lymphoproliferative diseases, including T-ALL and human T-cell leukemia virus type 1 (HTLV1)-associated adult T-cell leukemia (ATL) [233,234,235,236]. A synergistic growth inhibitory effect in NKTCL cell lines was also achieved by a combination of Ruxolitinib with the CDK4/6 inhibitor Ribociclib (LEE011) [237]. This indicates that Ruxolitinib might not only be efficient in myeloid malignancies, but could also be of therapeutic value for patients with lymphoid malignancies [100,238,239,240,241,242]. Clinical trials using Ruxolitinib are ongoing for treatment of different cancer entities, such as HTLV1-associated ATL (NCT01712659), relapsed/refractory ETP-ALL (NCT03613428), relapsed/refractory NK-cell or peripheral T-cell non-Hodgkin Lymphoma (NCT01431209, NCT02974647), and ALL (NCT03117751). Similarly, the inhibitory effects of Itacitinib on T-cell proliferation [226] may be repurposed to block transformed T-cell growth. Indeed, there is a clinical trial ongoing for the use of Itacitinib in combination with Alemtuzumab (anti-CD52) in T-PLL patients (NCT03989466) [243]. A therapeutic window remains to be determined at which Itacitinib exerts anti-tumor effects, while avoiding counterproductive immunosuppressive effects.

As described above, Tofacitinib potently inhibits T and NK cells in a time and dose-dependent manner. Therefore, it might be an option to treat malignancies derived from innate and adaptive lymphocytes. As indicated in Table 1, BaF3 cells expressing JAK3 mutations as well as JAK3-mutant T-ALL and NKTCL in vivo models are sensitive to Tofacitinib treatment [44,53,82]. Similarly, NKTCL cell lines with constitutive activation of JAK-STAT signaling, partially associated with JAK3 GOF mutations, respond to Tofacitinib [45,50,53,100,244]. A phase 2 clinical trial currently includes patients with relapsed/refractory NKTCL that are treated with a combination of Tofacitinib and the histone deacetylase inhibitor Chidamide (Table 2). First case reports demonstrated a moderate activity of the combined use of Tofacitinib and Ruxolitinib in T-PLL [243,245,246]. Tofacitinib inhibits STAT5-regulated miR-21 expression in cutaneous T-cell lymphoma (CTCL) and thereby blocks anti-apoptotic effects of miR-21 in malignant T cells [238,247]. Combination of the BCL2 inhibitor Venetoclax with Tofacitinib induced therapeutic responses in some hematological patients with relapsed/refractory T-ALL with surface IL-7R expression or IL-7R-pathway mutations and BCL2 expression [248]. In contrast, Tofacitinib was not effective in a patient with DDX3X-MLLT10 T-ALL carrying an activating JAK3 mutation [249]. Therefore, the individual genetic make-up of a hematological malignancy might determine its responsiveness to Tofacitinib treatment.

Tofacitinib is a promising therapeutic strategy also for large granular lymphocytic leukemia (LGLL)—a rare subtype of mature T- and NK-cell neoplasms that is characterized by clonal expansion of cytotoxic NK or T lymphocytes. LGLL has a largely indolent course and is frequently associated with autoimmune disorders [250,251,252,253]. T-cell LGLL (T-LGLL) in particular is associated with rheumatoid arthritis in 10–30% of the patients [254,255,256]. Other autoimmune disorders associated with LGLL are autoimmune-related cytopenia including neutropenia [251,253,257]. Deregulation of pro-survival pathways, including a deregulated JAK-STAT signaling, have been implicated in disease pathogenesis and support the expansion of auto-reactive lymphocytes [59,250,251,252,253]. T-LGLL shows a high proportion of somatic STAT3 GOF mutations [83,84,85,106]. Based on its immunosuppressive activity, Tofacitinib was studied as a salvage therapy for highly refractory T-LGLL with RA, showing encouraging response rates with improvement of RA and cytopenia symptoms and limited side effects [87]. Tofacitinib selectively induces apoptosis in STAT3-mutant T-LGLL cells compared to healthy CD8+ T cells [87]. There is hope that the suppressive effects of Ruxolitinib, Tofacitinib, and potentially Itacitinib on cytotoxic lymphocytes will be repurposed as novel therapies in NK/T-cell malignancies with Tofacitinib holding promise for specific types of leukemias associated with autoimmune symptoms [258].

## 5. Conclusions and Future Perspectives

Introduction of JAKinibs into the clinics has revolutionized treatment for MPN patients and patients suffering from autoimmune diseases and represents potential strategies for further diseases. The successful use of JAKinibs in autoimmune diseases highlights their potent immune suppressive effects, which if too broad, are of potential harm for the patient. This is particularly true for JAK1 inhibitors that block γc-dependent cytokines and may provoke severe immunological consequences, as summarized in Figure 1. Despite greater specificity of new generation inhibitors, such as Fedratinib and Pacritinib for JAK2, it is obvious that also these drugs interfere with the immune system. Both inhibitors have off-target effects on other kinases that contribute to immune suppression. It is currently impossible to untwine the beneficial anti-tumor effects from immune suppressive effects of JAKinibs. Novel inhibitors specifically targeting GOF mutations represent a potential route of innovation. One example is ZT55, which has been identified as a JAK2-inhibitor with increased affinity to *JAK2^V617F^* [259]. Of note, the immune suppressive potential of JAKinibs is explored in clinical trials for severe cases of COVID-19 with the aim to suppress pathologically enhanced IL-6 responses while keeping anti-viral immunity untouched. Among studied JAKinibs, JAK2-“specific” inhibitors represent the most promising option for these patients [260]. Another strategy to improve the effects of JAKinib treatment for patients is their use in combination therapies. In contrast to non-transformed NK cells, malignant NK cells are sensitive to the BCL2 inhibitor Venetoclax, which could be exploited to obtain a synergistic effect by combining with Ruxolitinib [100]. Synergistic effects provide an opportunity for reducing the dose of JAKinibs which would minimize undesired immunosuppressive effects. Alternatively, combinations of JAKinibs with immunomodulatory drugs may be tested as a strategy to counteract the immunosuppressive effects [261]. Novel compounds with high specificity or efficient combination therapies together with a robust immune profiling of JAKinibs are required in the future to provide the best benefit for JAKinib-treated patients.

## Figures and Tables

**Figure 1 cancers-13-02611-f001:**
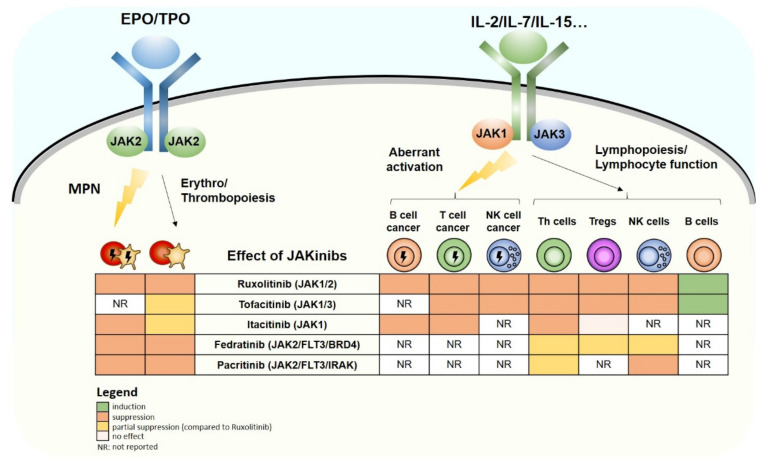
Schematic view of anti-tumor and immunosuppressive effects of JAKinibs downstream of JAK2/JAK2 and JAK1/JAK3-dependent pathways. Left side: In a physiological situation, EPO (erythropoietin) and TPO (thrombopoietin) signal via JAK2/JAK2 pairs to induce erythro-/thrombopoiesis (right side of the receptor). In a cancerous situation (indicated by 

), the same signaling pathway drives MPNs (myeloproliferative neoplasms; left side of the receptor). Right side: In a physiological situation, common gamma chain-dependent cytokines (e.g., IL-2/7/15) signal via JAK1/JAK3 pairs to induce lymphopoiesis and regulate T/B/NK-cell function (right side of the receptor). In a cancerous situation (indicated by 

), the same signaling pathway drives leukemias and lymphomas originating from T/B/NK cells (left side of the receptor). The consequences of JAKinib treatment on physiological and pathological effects of the pathways are indicated by colors (see legend).

**Table 1 cancers-13-02611-t001:** Overview on JAK-STAT gain-of-function (GOF) mutations in hematological malignancies.

JAK/ STAT	Type of mutations	Hematological malignancies ^1^	Examples for JAKinib sensitivity	References
JAK1	somatic GOF(e.g., *JAK1^S646F^, JAK1^S646P^, JAK1^V658I^*)	ALL (B-ALL, ETP-ALL, adult T-ALL), T-PLL, BIA-ALCL, ALK- ALCL, AML, MPN-unclassifiable, CMML	Ba/F3 cells expressing *JAK1^S646F^*, *JAK1^S646P^*, or *JAK1^V658I^* are sensitive to JAKinibs, including Ruxolitinib.	[4,15,39,40,41,42,51,74,75,76,77,78]
JAK2	germline GOF(e.g., *JAK2^R564Q^, JAK2^V617I^*)	hereditary essential thrombocytosis	Ba/F3-MPL cells expressing *JAK2^R564Q^* are more sensitive to Ruxolitinib than *JAK2^V617F^*-expressers.	[36,37,38]
	somatic GOF(e.g., *JAK2^V617F^*)	MPN (PV > ET, PMF), AML, pediatric, and DS-ALL	Ruxolitinib is approved for treatment of MPN.	[26,27,28,29,30,31,32,33,34,35]
JAK3	germline GOF(*JAK3^Q507P^*)	familial CLPD-NK		[79]
	somatic GOF(e.g., *JAK3^M511I^, JAK3^A572V^, JAK3^A573V^*)	ALL (ETP-ALL, T-ALL), ATLL, T-PLL, AML, (DS-)AMKL, NKTCL	Ba/F3 cells expressing *JAK3^M511I^* or *JAK3^A573V^* are sensitive to Tofacitinib. *JAK3^A573V^* mutant NKTCL, and *JAK3^M511I^* mutant T-ALL-like disease models are responsive to Tofacitinib treatment in vivo.	[4,15,41,43,44,45,46,47,48,49,50,51,52,53,80,81,82]
TYK2	germline GOF(*TYK2^G716V^, TYK2^P760L^*)	pediatric ALL (B-ALL, T-ALL)		[55,56]
	somatic GOF(e.g., *TYK2^E957D^*)	T-ALL	Ba/F3 cells expressing *TYK2^E957D^* are sensitive to JAK inhibitor I.	[54,56]
STAT3	germline GOF(e.g., *STAT3^K392R^*)	pediatric LGLL		[64]
	somatic GOF(e.g., *STAT3^Y640F^, STAT3^D661Y/I/V/H^*)	T-LGLL, NK-LGLL, ALK-ALCL, HSTL, DLBCL NOS, NKTCL, CLPD-NK, ANKL, Sezary syndrome	Tofacitinib could be a promising salvage therapy for refractory T-LGLL patients with or without STAT3 mutations.	[60,61,63,65,83,84,85,86,87,88]
STAT5B	somatic GOF(e.g., *STAT5B^N642H^, STAT5B^Y665H/F^*)	NKTCL, ANKL, NK-LGLL, T-LGLL, T-PLL, T-ALL, MEITL, HSTL, PCTL, Sezary Syndrome, PTCL-NOS, AML, AAA, CNL, Eosinophilia	*STAT5B^N642H^*-driven CD8+ T-cell disease and CD56+ T-LGL (NKT) leukemia models are sensitive to Ruxolitinib. *STAT5B^N642H^* T-ALL is sensitive to JAK1/JAK3 inhibitors.	[59,60,61,62,72,73,86,89,90,91,92,93,94,95,96,97,98,99,100,101,102,103,104,105,106,107,108,109]
STAT6	somatic GOF(e.g., *STAT6^D419H/G/A^*)	CHL, FL, RR-DLBCL, PMBCL	*STAT6^D419^*-missense mutated PMBCL cell lines are sensitive to JAK2 inhibitors.	[66,67,68,69]

^1^ Abbreviations: AAA—acquired aplastic anemia; ALK-ALCL—anaplastic lymphoma kinase negative anaplastic large cell lymphoma; ALL—acute lymphoblastic leukemia; AML—acute myeloid leukemia; ANKL—aggressive natural killer cell leukemia; ATLL—adult T-cell leukemia lymphoma; B-ALL—B-cell acute lymphoblastic leukemia; BIA-ALCL—breast implant-associated anaplastic large cell lymphoma; CHL—classical hodgkin lymphoma; CLPD-NK—chronic lymphoproliferative disorders of natural killer cells; CMML—chronic myelomonocytic leukemia; CNL—chronic neutrophilic leukemia; DLBCL NOS—diffuse large B-cell lymphoma, not-otherwise-specified; DS-ALL—down syndrome acute lymphoblastic leukemia; (DS)-AMKL—(down syndrome) acute megakaryoblastic leukemia; ET—essential thrombocythemia; ETP-ALL—early T-cell precursor ALL; FL—follicular lymphoma; HSTL—hepatosplenic T-cell lymphoma; LGLL—large granular lymphocytic leukemia; MEITL—monomorphic epitheliotropic intestinal T cell lymphoma; MPN—myeloproliferative neoplasm; NK-LGLL—natural killer cell large granular lymphocytic leukemia; NKTCL—extranodal NK/T-cell lymphoma; PMBCL—primary mediastinal B-cell lymphoma; PMF—primary myelofibrosis; PCTL—primary cutaneous γδ T-cell lymphoma; PTCL-NOS—peripheral T-cell lymphoma not-other-specified; PV—polycythemia vera; RR-DLBCL—relapsed-refractory diffuse large B-cell lymphoma; T-ALL—T-cell acute lymphoblastic leukemia; T-LGLL—T-cell large granular lymphocytic leukemia; T-PLL—T-cell prolymphocytic leukemia.

**Table 2 cancers-13-02611-t002:** Ongoing clinical trials for JAKinibs in hematological malignancies (as for April 2021).

NCT number	JAKinib	Phase	Disease(s) ^2^
NCT02723994	Ruxolitinib (JAK1/2)	2	ALL
NCT03571321		1	ALL (Ph-like)
NCT03874052		1	AML
NCT03286530		2	AML
NCT04055844		2	AML, MDS
NCT03654768		2	CML
NCT03610971		2	CML (chronic phase)
NCT03722407		2	CMML
NCT03801434		2	Eosinphilic syndromes
NCT04669210		2	GvHD, HSCT complications, ALL, AML
NCT02613598		1	(Non)Hodgkin Lymphoma
NCT03681561		1/2	Hodgkin Lymphoma
NCT03017820		1	Leukemia/lymphoma
NCT03878199		1/2	MPN
NCT04281498		2	MPN
NCT04041050		1	MPN
NCT02158858		1/2	MPN, MDS
NCT03558607		1/2	sAML
NCT01712659		1/2	T-cell Leukemia
NCT03613428		1/2	T-cell Leukemia
NCT03117751		2/3	T-cell Leukemia/Lymphoma
NCT02974647		2	T/NK lymphoma
NCT04282187	Ruxolitinib (JAK1/2) or Fedratinib (JAK2)	2	MPN, AML
NCT04282187	Fedratinib (JAK2)		MPN, AML
NCT03598959	Tofacitinib (JAK1/3)	2	T/NK lymphoma
NCT04640025	Itacitinib (JAK1)	2	MF
NCT01633372/NCT04629508		2	MPN
NCT03144687		2	MPN
NCT04061421		1/2	MDS/MPN
NCT03697408		1/2	classical HL
NCT02760485		1/2	Relapsed or Refractory DLBCL
NCT01905813		1	B-cell Malignancies (previously treated)
NCT02018861/NCT04509700		1/2	B-cell Malignancies (previously treated)
NCT03989466		1	(recurrent) T-PLL
NCT04173494	Momelotinib (JAK1/2)	3	MPN (pMF, PV)
NCT03645824	Patricinib (JAK2)	2	MF
NCT03165734		3	MF (primary and secondary)
NCT02891603		1/2	GvHD

^2^ Abbreviations: ALL—Acute lymphoblastic leukemia; (s)AML—(secondary) acute myeloid leukemia; CML—Chronic myeloid leukemia; CMML—Chronic myelomonocytic leukemia; DLBCL—Diffuse Large B-Cell Lymphoma; GvHD—graft versus host disease; HL—Hodgkin lymphoma; HSCT—hematopoietic stem cell transplant; MDS—Myelodysplastic syndrome; MPN—myeloproliferative neoplasm; Ph—Philadelphia chromosome; (p)MF—(primary) myelofibrosis; T-PLL—T-cell-prolymphocytic leukemia. Search strategy: clinicaltrials.gov were searched for trials fulfilling following criteria: condition or disease—Leukemia/Lymphoma; other terms—name of JAKinib; excluded: terminated, withdrawn, suspended, or status unknown.

**Table 3 cancers-13-02611-t003:** Mutations in the JAK-STAT pathway resulting in patient’s immune dysfunctions and immunological phenotypes of respective knock-out/-in mice.

JAK/ STAT	Type of mutations	Immune phenotype of patients ^3^	Immune phenotype of knockout/-in mice	References
JAK1	LOF (e.g., *JAK1^P733L^; JAK1^P832S^*)	Immunodeficiency (early onset cancer and recurrent mycobacterial infections)	Perinatally lethal; severe reduction of pre–B cells, and mature T and B lymphocytes	[125,126]
JAK3	LOF (e.g., *JAK3^Y100C^; JAK3^D169E^*)	autosomal recessive T-B+NK- SCID (null mutations), broader range of clinical immunosuppressive phenotypes	Defective T, B, ILC (incl. NK) cell development	[80,127,128,129,130,131,132,133]
TYK2	LOF (e.g., *TYK2^I684S^*)	Mycobacterial and viral infections	Impaired T and NK-mediated anti-viral, anti-bacterial, and anti-tumor responses	[123,124,134,135,136,137,138]
STAT1	LOF (e.g., *STAT1^K201L^; STAT1^K211R^*)	Complete deficiency: mycobacteria, virus infection; dysfunctional NK cells; partial deficiency: mycobacteria but no virus infection	Impaired responses to Type I and Type II IFN, increased susceptibility to infections, impaired NK cells	[139,140,141,142,143,144,145]
	GOF (e.g., *STAT1^Y170N^; STAT1^C174R^*)	viral, bacterial infections, combined immunodeficiency (reduced memory B, Th17 cells, impaired NK cells); autoinflammation, organ-specific autoimmune disorders	Impaired IL-17 immunity	[146,147,148,149,150]
STAT2	LOF (e.g., *STAT2^c.1836C4A^*)	primary immunodeficiency (viral infections)	Impaired response to Type I IFN and susceptibility to viral infections	[113,114,151,152]
STAT3	LOF (e.g., *STAT3^Y657N^*)	AD-HIES, primary immunodeficiency (susceptibilities to infections, impaired Th17 and B cells)	Embryonically lethal	[110,153,154,155]
	GOF (e.g., *STAT3^Q344H^*)	Immune deficiency (reduced memory B cells, NK cells, pDCs); various organ autoimmunity	-	[64,110,156]
STAT4	LOF (e.g., *STAT4^E651V^*)	Fungal infections	Inhibited Th1 differentiation	[115,116,157]
STAT5B	LOF (e.g., *STAT5B^A630P^*)	combined immunodeficiency (Treg deficiency, reduced T cells and NK cells) serve viral infections; autoimmune symptoms	Impaired NK and T cells	[158,159,160,161,162,163,164]

**^3^** Abbreviations: AD-HIES—autosomal dominant hyper-IgE syndrome; IFN—interferon; ILC—innate lymphocyte; NK—natural killer; pDC—plasmacytoid dendritic cells; SCID—severe combined immunodeficiency; Treg—T regulatory cells.

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
