# Peer review of "Untwining Anti-Tumor and Immunosuppressive Effects of JAK Inhibitors—A Strategy for Hematological Malignancies?"

_cancers, 2021, doi:10.3390/cancers13112611_

Round 1

Reviewer 1 Report

The review titled "Untwining the Anti-tumor and Immunosuppressive Effects of Janus Kinase Inhibitors" effectively summarizes JAK-STAT signaling and associated mutations that occur in hematological malignancies, therapeutic strategies involving JAKinibs that are currently being tested in clinical trials, and JAKinibs that are currently being tested for anti-cancer activity in hematological malignancies.

The review is thorough and it is evident that a lot of effort has gone in summarizing the trials and trial-related data for this review. The review is well structured. I appreciate the inclusion of GVHD and venetoclax combination with ruxolitinib. 

Comments:

  1. While baricitinib is not currently being tested in a clinical trial setting to treat any hematological cancer, in my opinion, it is worth mentioning that occurrence of lymphoma is one of the serious adverse event albeit rare with the use of baricitinib (PMID: 30219772 is a good source).
  2. In conclusions and future perspectives, I would like to see some discussion (whatever fits within the scope of the article even if it is preclinical evidence) with potential combination therapies for treating malignancies. JAKinibs as monotherapy, for eg., ruxolitinib in MPN diminishes symptom burden but does not affect the disease or clonal landscape. 
  3. Lastly, I would suggest modifying the title of this review to reflect its content better. I would suggest something along the lines of  "Untwining the Anti-tumor and Immunosuppressive Effects of Janus Kinase Inhibitors- A Strategy to Effectively Target Hematological Malignancies?"

Author Response

Reviewer 1

The review titled "Untwining the Anti-tumor and Immunosuppressive Effects of Janus Kinase Inhibitors" effectively summarizes JAK-STAT signaling and associated mutations that occur in hematological malignancies, therapeutic strategies involving JAKinibs that are currently being tested in clinical trials, and JAKinibs that are currently being tested for anti-cancer activity in hematological malignancies.

The review is thorough and it is evident that a lot of effort has gone in summarizing the trials and trial-related data for this review. The review is well structured. I appreciate the inclusion of GVHD and venetoclax combination with ruxolitinib

We are very grateful for the positive feedback of the reviewer.

Comments:

1. While baricitinib is not currently being tested in a clinical trial setting to treat any hematological cancer, in my opinion, it is worth mentioning that occurrence of lymphoma is one of the serious adverse event albeit rare with the use of baricitinib (PMID: 30219772 is a good source).

We thank the reviewer for this important remark. As our focus is inhibitors being tested in clinical trials for hematological malignancies we have included a short comment in the Ruxolitinib part (line 218-220) and hope that the reviewer will find that appropriate.

2. In conclusions and future perspectives, I would like to see some discussion (whatever fits within the scope of the article even if it is preclinical evidence) with potential combination therapies for treating malignancies. JAKinibs as monotherapy, for eg., ruxolitinib in MPN diminishes symptom burden but does not affect the disease or clonal landscape

We thank the reviewer for pointing out this current topic. We have included a paragraph about combination therapy in Conclusions and future perspectives (line 454-462). We hope for the understanding of the reviewer that a detailed description is beyond the scope of this review.

3. Lastly, I would suggest modifying the title of this review to reflect its content better. I would suggest something along the lines of "Untwining the Anti-tumor and Immunosuppressive Effects of Janus Kinase Inhibitors- A Strategy to Effectively Target Hematological Malignancies?"

We greatly appreciate the suggestion to improve the title of our manuscript. We have implemented this change, choosing following new title for the manuscript: Untwining Anti-tumor and Immunosuppressive Effects of JAK Inhibitors – A Strategy for Hematological Malignancies?

Reviewer 2 Report

The authors excellently reviewed and summarized current knowledge concerning inhibitors against JAK-STAT signaling system and their potentials of immunomodulatory and anti-tumor effects for lymphoid cells.

  1. On Table 3, immunophenotypes of knockout mice were mentioned on each molecule. It might be useful for the readers to include information of knock-in mice of their mutants and immune status.
  2. There was no mention on basic and clinical effects of ruxolitinib for graft-versus-host disease (GVHD), while other agents such as itacitinib and its clinical trial on GVHD were presented.
  3. It is necessary to include the methods to access and find information of clinical trails of JAK inhibitors on Table 2.

Author Response

Reviewer 2

The authors excellently reviewed and summarized current knowledge concerning inhibitors against JAK-STAT signaling system and their potentials of immunomodulatory and anti-tumor effects for lymphoid cells.

 We are very grateful for the positive feedback of the reviewer.

1. On Table 3, immunophenotypes of knockout mice were mentioned on each molecule. It might be useful for the readers to include information of knock-in mice of their mutants and immune status.

 We appreciate the comment of the reviewer and agree that adding the immune effects of mutant knock-in mice of JAK-STAT pathway members could be of interest to the readers. The well described JAK2V617F (Li et al., 2011) or STAT5BN642H (Pham et al., 2018) mutant mice develop leukemia very early and the direct effects of the mutation on the immune system are hard to be distinguished from any secondary effects caused by the malignancy and the tumor environment. After intensive discussion, we reached the conclusion that immune phenotypes of mutant mice developing leukemia are very complex and their inclusion is therefore beyond the focus of this review. Importantly, in Table 3 we have included the GOF-Stat1 knock-in (GOF-Stat1R274Q) mouse which mimics human immunodeficiency (Tamaura et al., 2020).

2. There was no mention on basic and clinical effects of ruxolitinib for graft-versus-host disease (GVHD), while other agents such as itacitinib and its clinical trial on GVHD were presented.

We apologize for the inconsistencies regarding the GvHD information. We have now included a basic information about the effects of Ruxolitinib in GvHD (line 207-209; 241-242).

3. It is necessary to include the methods to access and find information of clinical trails of JAK inhibitors on Table 2.

We thank the reviewer for suggesting to add this important information. We included the details of our search strategy in the foot note of Table 2 (line 152-154).

References

Li, J.; Kent, D. G.; Chen, E.; Green, A. R. Mouse Models of Myeloproliferative Neoplasms: JAK of All Grades. DMM Disease Models and Mechanisms. The Company of Biologists May 1, 2011, pp 311–317. https://doi.org/10.1242/dmm.006817.

Pham, H. T. T.; Maurer, B.; Prchal-Murphy, M.; Grausenburger, R.; Grundschober, E.; Javaheri, T.; Nivarthi, H.; Boersma, A.; Kolbe, T.; Elabd, M.; et al. STAT5B N642H Is a Driver Mutation for T Cell Neoplasia. J. Clin. Invest., 2018, 128 (1), 387–401. https://doi.org/10.1172/JCI94509.

Tamaura, M.; Satoh-Takayama, N.; Tsumura, M.; Sasaki, T.; Goda, S.; Kageyama, T.; Hayakawa, S.; Kimura, S.; Asano, T.; Nakayama, M.; et al. Human Gain-of-Function STAT1 Mutation Disturbs IL-17 Immunity in Mice. Int. Immunol., 2020, 32 (4), 259–272. https://doi.org/10.1093/intimm/dxz079